# Design and Characterisation of a Fast Steering Mirror Compensation System Based on Double Porro Prisms by a Screw-Ray Tracing Method

**DOI:** 10.3390/s18114046

**Published:** 2018-11-20

**Authors:** Yu-Hao Chang, Chien-Sheng Liu, Chih-Chun Cheng

**Affiliations:** 1Department of Mechanical Engineering, National Chung Cheng University, Chiayi County 62102, Taiwan; tako79716@gmail.com (Y.-H.C.); imeccc@ccu.edu.tw (C.-C.C.); 2Advanced Institute of Manufacturing with High-Tech Innovations, National Chung Cheng University, Chiayi County 62102, Taiwan; 3Department of Mechanical Engineering, National Cheng Kung University, Tainan City 70101, Taiwan

**Keywords:** fast steer mirror, FSM, Double Porro prisms, laser drift, skew-ray tracing method

## Abstract

This study proposes a novel FSM compensation system for four degrees of freedom (DOF) laser errors compensation, which has the advantage of shorter optical path length, fewer elements and an easier set-up process, meaning that it can be used at different locations. A commercial software, Zemax, is used to evaluate the function of the proposed FSM compensation system and the mathematical modelling of the proposed FSM compensation system is established by using a skew-ray tracing method. Finally, the proposed FSM compensation system is then verified experimentally using a laboratory-built prototype and the result shows that the proposed FSM compensation system achieves the ability to compensate the 4 DOF of the laser source.

## 1. Introduction

Laser technology is widely utilised in different fields, such as engineering inspection, communication industry, semiconductor industry, and military industry [1,2,3]. Moreover, the stability of the laser device will directly influence the performance and the price of the equipment. However, the laser light is not a stable light source and it will ramble and shift as the time varies [4,5,6,7]. There are certain studies which mention that the intensity fluctuations of the laser source are related to the input current [8], while the environmental conditions (e.g., atmospheric), vibrations and thermally-induced motion may cause the laser to misalign [9]. The laser source drift contains two DOF angular errors and two DOF displacement errors, although even a small angular error will be magnified by the optical path length and reduce the quality of the product [10,11]. Therefore, it is necessary to improve the laser point stability. In order to solve the laser point stability problem, there are different types of methods which can be used to reduce the laser drift. The passive compensation method reduces the laser drift by controlling the power or frequency fluctuations of the laser device [12,13]. This method has a lower cost but also a lower efficiency; it cannot correct the laser point instantly and cannot correct the drift caused by environmental factors. By contrast, the fast steer mirror (FSM) compensation system can correct the laser point actively via its closed-loop system, and so it has the advantage of being able to correct the drift caused by environmental factors and the laser device itself; moreover, it can also compensate for the laser error instantly and continuously. 

The FSM compensation system plays an important role in the free-space optical communications and auto adaptation optical system for controlling the laser beam direction, because it has good dynamic performance [14,15,16]. In order to correct the 4 DOF laser errors to achieve the best laser point stability, there are two FSMs and two sensors which are used to correct the two angular errors and two displacement errors in the conventional FSM compensation system [17]. However, the conventional FSM compensation system has the disadvantage of longer optical path length, which means that errors are magnified due to the system’s excessive elements. Therefore, this study proposes a new FSM compensation system that gives the existing compensation system the advantage of shorter optical path length, fewer elements, and easier set up at different locations. The proposed FSM compensation system is characterised using simulation methods and mathematical modelling, then evaluated by experimentally using a laboratory-built prototype.

## 2. Design of Proposed FSM Compensation System and Simulations 

As shown in Figure 1, a conventional FSM compensation system is constructed using two 2-axes FSMs, two beam splitters, and two PSDs. In the FSM compensation system, PSDs play the role of detecting the four DOF laser source errors and the FSM is represented as a mirror that is mounted over the actuators to steer the output laser beam and compensate for laser errors. In order to achieve better laser point stability, it is necessary to compensate for all four DOF errors of the laser point. Therefore, the conventional FSM compensation system needs to place two 2-axes FSMs to compensate for the two DOF angular errors and the two DOF displacement errors respectively. However, these two 2-axes FSMs will produce an extra optical path between themselves, which means that the laser point will involve more environmental disturbance. Moreover, and as mentioned in Section 1, even a small angular error will be magnified by the optical path length, and thus it is essential to reduce the optical path length of the optical system.

This study proposes a new FSM structure to reduce the optical path length of the conventional FSM compensation system. As shown in Figure 2, the new FSM structure modifies the two FSMs to a new 4-axes FSM structure, which removes the optical path between two FSMs directly and reduces the environmental disturbance. However, one normal mirror could produce 2 DOF steering only, and so the Double Porro prisms, which could produce 4-DOF steering are used in the new FSM structure. 

Double Porro prisms are variants of the 90° prisms that are used as a pair to displace and invert a beam; indeed, these are widely employed in binoculars [18,19,20] and as a beam rotator [21]. As shown in Figure 3, the 90° prism could tilt and shift the laser path when the 90° prism is rotated along the Y-axis and shifted along the Y-axis; as such, when a pair of 90° prisms are put together vertically (which are called Double Porro prisms), they would have the ability of 4 DOF steering. Therefore, the characteristic of the Double Porro prisms mentioned above is combined with the 4-DOF actuator to compensate for the laser error. 

In order to verify the function of the proposed FSM compensation system, a commercial software, Zemax, is used to build the optical path and perform a multi-point laser error analysis which is shown in Figure 4. The analysis assigns the 2 DOF displacement errors and 2 DOF angular errors from −1 mm to +1 mm and from −1° to +1°, respectively. Moreover, the distance and the angle between each interval are 0.1 mm and 0.1°, respectively. The distributions of $11^{4}$ laser points are shown in Figure 5. However, all errors could be compensated for by the Double Porro prisms through the function of optimisation, which is shown in Figure 6; moreover, the compensated error is under 0.01 μm. The result verifies that the proposed FSM compensation system has the ability to compensate for 4 DOF laser errors and there is no singularity in the system.

## 3. Proposed FSM Compensation Method

Firstly, the proposed FSM compensation system detects the laser drift by two PSDs and uses a mathematical algorithm to calculate the 2-DOF angular errors ($\theta_{x}$, $\theta_{y}$) and 2-DOF displacement errors ($\delta_{x}$, $\delta_{y}$), following which it calculates the compensation values (${\theta^{\prime }}_{x}$, ${\theta^{\prime }}_{y}$, ${\delta^{\prime }}_{x}$, ${\delta^{\prime }}_{y}$). Thereafter, the system drives the 4-axes actuator to compensate for the laser source errors.

### 3.1. Flat-Surface Skew-Ray Tracing Method

The flat-boundary skew-ray tracing method, proposed by Lin [18,22], is used in this study to build the mathematical algorithm of the proposed FSM compensation system. The transformation matrix ${}^{R}A_{i}$ is relative to the universal coordinate system (*R*) which is built on the boundary of each optical element, as shown below:(1)$${}^{R}A_{i}=\left[\begin{array}{cccc} I_{ix} & J_{ix} & K_{ix} & t_{ix}\\ I_{iy} & J_{iy} & K_{iy} & t_{iy}\\ I_{iz} & J_{iz} & K_{iz} & t_{iz}\\ 0 & 0 & 0 & 1 \end{array}\right]\;$$

The vectors ${\left[\begin{array}{cccc} I_{ix} & I_{iy} & I_{iz} & 0 \end{array}\right]}^{T}$, ${\left[\begin{array}{cccc} J_{ix} & J_{iy} & J_{iz} & 0 \end{array}\right]}^{T}$ and ${\left[\begin{array}{cccc} K_{ix} & K_{iy} & K_{iz} & 0 \end{array}\right]}^{T}$ describe the orientation of the three unit vectors of coordinate frame i with respect to another coordinate frame R, respectively. Vector ${\left[\begin{array}{cccc} t_{ix} & t_{iy} & t_{iz} & 0 \end{array}\right]}^{T}$ is the position vector of origin of coordinate frame i with respect to that of another coordinate frame R.

The laser ray is considered as unit direction vector in the flat-boundary skew-ray tracing method, as shown in Figure 7. $P_{i-1}$ and $\ell_{i-1}$ express the incident point and incident unit direction vector, while $P_{i}$ and $\ell_{i}$ express the point on the boundary and the reflected ray or refracted ray. They could be expressed as follows:(2)$$P_{i-1}={\left[\begin{array}{cccc} P_{i-1x} & P_{i-1y} & P_{i-1z} & 1 \end{array}\right]}^{T}\;$$
(3)$$\ell_{i-1}={\left[\begin{array}{cccc} \ell_{i-1x} & \ell_{i-1y} & \ell_{i-1z} & 1 \end{array}\right]}^{T}\;$$
(4)$$P_{i}={\left[\begin{array}{cccc} P_{ix} & P_{iy} & P_{iz} & 1 \end{array}\right]}^{T}={\left[\begin{array}{cccc} P_{i-1x}+\ell_{i-1x}\lambda_{i} & P_{i-1y}+\ell_{i-1y}\lambda_{i} & P_{i-1z}+\ell_{i-1z}\lambda_{i} & 1 \end{array}\right]}^{T}\;$$
$\lambda_{i}$ is the geometrical length from the incident point $P_{i-1}$ to the point on the boundary:(5)$$\lambda_{i}=\frac{-(I_{iz}P_{i-1x}+J_{iz}P_{i-1y}+K_{iz}P_{i-1z}+t_{iz})}{I_{iz}\ell_{i-1x}+J_{iz}\ell_{i-1y}+K_{iz}\ell_{i-1z}}=\frac{-B_{i}}{G_{i}}\;$$

According to Snell’s Law, when the laser incidents upon a flat surface and is reflected, the unit direction vector $\ell_{i}$ could be expressed as follows:(6)$$\ell_{i}={\left[\begin{array}{cccc} \ell_{ix} & \ell_{iy} & \ell_{iz} & 0 \end{array}\right]}^{T}={\left[\begin{array}{cccc} \ell_{i-1x}-2I_{iz}G_{i}n_{ix} & \ell_{i-1y}-2J_{iz}G_{i}n_{iy} & \ell_{i-1z}-2K_{iz}G_{i}n_{iz} & 0 \end{array}\right]}^{T}\;$$

If the laser incident is refracted, the unit direction vector $\ell_{i}$ could be expressed as follows:(7)$$\ell_{i}=\left[\begin{array}{c} \ell_{ix}\\ \ell_{iy}\\ \ell_{iz}\\ 0 \end{array}\right]=\left[\begin{array}{c} -n_{ix}\sqrt{1-N_{i}^{2}+{\left(N_{i}C\theta_{i}\right)}^{2}}+N_{i}(\ell_{i-1x}+n_{ix}C\theta_{i})\\ -n_{iy}\sqrt{1-N_{i}^{2}+{\left(N_{i}C\theta_{i}\right)}^{2}}+N_{i}(\ell_{i-1y}+n_{iy}C\theta_{i})\\ -n_{iz}\sqrt{1-N_{i}^{2}+{\left(N_{i}C\theta_{i}\right)}^{2}}+N_{i}(\ell_{i-1z}+n_{iz}C\theta_{i})\\ 0 \end{array}\right]\;$$

Note that $n_{i}$ is normal vector and $N_{i}$ is the index of refraction defined by Snell’s Law [23,24].

### 3.2. Laser Drift Errors Measurement

The proposed FSM compensation system includes two optical paths: the optical path Ι, which is from the laser source to PSD 1, and the optical path Π to PSD 2, as shown in Figure 8. In order to build the full skew-ray tracing of the proposed FSM compensation system, in the beginning coordinate system, the homogeneous coordinate transformation matrix of each boundary, incident point, and incident unit direction vector should be defined. As shown, the reference coordinate system ${\left(xyz\right)}_{0}$ is built on the laser source, while the coordinate ${\left(xyz\right)}_{i}$ is built on the boundary of optical elements; moreover, the parameters of the homogeneous coordinate transformation matrix of the coordinate system ${\left(xyz\right)}_{i}$ relative to ${\left(xyz\right)}_{i+1}$ are shown in Table 1. $L_{i}$ is the straight-line distance between each coordinate system.

The incident point and incident unit direct vector of the laser source relative to the reference coordinate system are defined as:(8)$$P_{0}={\left[\begin{array}{cccc} 0 & 0 & 0 & 1 \end{array}\right]}^{T},\;\ell_{0}={\left[\begin{array}{cccc} 0 & 0 & 1 & 0 \end{array}\right]}^{T}\;$$

A transformation matrix $T_{1}$ is used to distinguish between 2-DOF angular errors ($\theta_{x}$, $\theta_{y}$) and 2-DOF displacement errors ($\delta_{x}$, $\delta_{y}$) of laser drift:(9)$$T_{1}=\left[\begin{array}{cccc} 1 & 0 & 0 & \delta_{X}\\ 0 & 1 & 0 & \delta_{Y}\\ 0 & 0 & 1 & 0\\ 0 & 0 & 0 & 1 \end{array}\right]\left[\begin{array}{cccc} 1 & 0 & 0 & 0\\ 0 & \cos(\theta_{X}) & -\sin(\theta_{X}) & 0\\ 0 & \sin(\theta_{X}) & \cos(\theta_{X}) & 0\\ 0 & 0 & 0 & 1 \end{array}\right]\left[\begin{array}{cccc} \cos(\theta_{y}) & 0 & \sin(\theta_{y}) & 0\\ 0 & 1 & 0 & 0\\ -\sin(\theta_{y}) & 0 & \cos(\theta_{y}) & 0\\ 0 & 0 & 0 & 0 \end{array}\right]\;$$

Furthermore, the laser source containing 2-DOF angular errors ($\theta_{x}$, $\theta_{y}$) and 2-DOF displacement error ($\delta_{x}$, $\delta_{y}$) could be written as:(10)$$P_{0}\prime =T_{1}P_{0},\;\ell_{0}\prime =T_{1}\ell_{0}\;$$

The laser ray on the coordinate ${\left(xyz\right)}_{0}$ would incident to the first optical boundary ${\left(xyz\right)}_{1}$, then reflect to the next optical boundary ${\left(xyz\right)}_{2}$. The incident point ${}^{1}P_{1}$ and the reflected ray ${}^{1}\ell_{1}$ on the first optical boundary, which are relative to coordinate ${\left(xyz\right)}_{1}$, could be calculated by using Equation (4) to Equation (6) and the parameter of the homogeneous coordinate transformation matrix ${}^{1}A_{0}$ of the coordinate system ${\left(xyz\right)}_{0}$ relative to ${\left(xyz\right)}_{1}$, as shown in Table 1. Now the ${}^{1}P_{1}$ and ${}^{1}\ell_{1}$ are considered as the incident point and the incident unit direct vector of the next optical boundary ${\left(xyz\right)}_{2}$; as such, the same method mentioned above is then used to obtain the incident points ${}^{2}P_{2}$ and ${}^{2}\ell_{2}$ on the second optical boundary, which are relative to coordinate ${\left(xyz\right)}_{2}$. Sequentially, the incident point and the incident unit direct vector on each optical boundary could be obtained. Finally, the spot coordinates on PSD 1 and PSD 2 could be obtained as simultaneous higher order equations in four variables as follows:(11)$$P_{PSD1}=\left[\begin{array}{cccc} X_{1}\left(\theta_{x},\theta_{y},\delta_{x},\delta_{y}\right) & Y_{1}\left(\theta_{x},\theta_{y},\delta_{x},\delta_{y}\right) & 0 & 1 \end{array}\right]\;$$
(12)$$P_{PSD2}=\left[\begin{array}{cccc} X_{2}\left(\theta_{x},\theta_{y},\delta_{x},\delta_{y}\right) & Y_{2}\left(\theta_{x},\theta_{y},\delta_{x},\delta_{y}\right) & 0 & 1 \end{array}\right]\;$$

After the whole flat-boundary skew-ray tracing, the relationship between PSD 1, PSD 2, and the four laser drift errors is built as Equations (11) and (12), where, $X_{1}$, $X_{2}$, $Y_{1}$ and $Y_{2}$ are the coordinate values of X and Y direction on PSD 1 and PSD 2, respectively. The four laser drift errors could be obtained after solving Equations (11) and (12):(13)$$\theta_{x}=f_{1}\left(X_{1},X_{2},Y_{1},Y_{2}\right)\;$$
(14)$$\theta_{y}=f_{2}\left(X_{1},X_{2},Y_{1},Y_{2}\right)\;$$
(15)$$\delta_{x}=f_{3}\left(X_{1},X_{2},Y_{1},Y_{2}\right)\;$$
(16)$$\delta_{y}=f_{4}\left(X_{1},X_{2},Y_{1},Y_{2}\right)\;$$

### 3.3. Laser Drift Errors Compensation

This section builds a new coordinate system to calculate the compensation values of the Double Porro prisms, which is shown in Figure 9. The parameters of the homogeneous coordinate transformation matrix of the coordinate system ${\left(xyz\right)}_{i}$ relative to ${\left(xyz\right)}_{i+1}$ are shown in Table 2.

The coordinate system ${\left(xyz\right)}_{1}$ and ${\left(xyz\right)}_{6}$ are set under the Double Porro prisms, which simulate the motion of the 4-axes actuator. The incident point and incident unit direct vector of a laser source that contains 4-DOF errors could be written as: (17)$$P_{0}\prime =T_{1}P_{0},\;\ell_{0}\prime =T_{1}\ell_{0}\;$$

Note that the angular errors ($\theta_{x}$, $\theta_{y}$) and displacement errors ($\delta_{x}$, $\delta_{y}$) are known to be constant in this sub-section. The laser ray on the coordinate ${\left(xyz\right)}_{0}$ would incident to the first virtual boundary ${\left(xyz\right)}_{1}$, then refract to the next optical boundary ${\left(xyz\right)}_{2}$. The transformation matrices $T_{2}$ and $T_{3}$ are used to describe 2-DOF angular motion ($\theta_{x}^{\prime }$, $\theta_{y}^{\prime }$) and 2-DOF displacement motion ($\delta_{x}^{\prime }$, $\delta_{y}^{\prime }$) of Double Porro prisms:(18)$$T_{2}=\left[\begin{array}{cccc} 1 & 0 & 0 & \delta_{x}^{\prime }\\ 0 & 1 & 0 & \delta_{y}^{\prime }\\ 0 & 0 & 1 & 0\\ 0 & 0 & 0 & 1 \end{array}\right]\left[\begin{array}{cccc} 1 & 0 & 0 & 0\\ 0 & \cos(\theta_{x}^{\prime }) & -\sin(\theta_{x}^{\prime }) & 0\\ 0 & \sin(\theta_{x}^{\prime }) & \cos(\theta_{x}^{\prime }) & 0\\ 0 & 0 & 0 & 1 \end{array}\right]\left[\begin{array}{cccc} \cos(\theta_{y}^{\prime }) & 0 & \sin(\theta_{y}^{\prime }) & 0\\ 0 & 1 & 0 & 0\\ -\sin(\theta_{y}^{\prime }) & 0 & \cos(\theta_{y}^{\prime }) & 0\\ 0 & 0 & 0 & 0 \end{array}\right]\;T_{3}=\left[\begin{array}{cccc} 1 & 0 & 0 & -\delta_{x}^{\prime }\\ 0 & 1 & 0 & -\delta_{y}^{\prime }\\ 0 & 0 & 1 & 0\\ 0 & 0 & 0 & 1 \end{array}\right]\left[\begin{array}{cccc} 1 & 0 & 0 & 0\\ 0 & \cos(-\theta_{x}^{\prime }) & -\sin(-\theta_{x}^{\prime }) & 0\\ 0 & \sin(-\theta_{x}^{\prime }) & \cos(-\theta_{x}^{\prime }) & 0\\ 0 & 0 & 0 & 1 \end{array}\right]\left[\begin{array}{cccc} \cos(-\theta_{y}^{\prime }) & 0 & \sin(-\theta_{y}^{\prime }) & 0\\ 0 & 1 & 0 & 0\\ -\sin(-\theta_{y}^{\prime }) & 0 & \cos(-\theta_{y}^{\prime }) & 0\\ 0 & 0 & 0 & 0 \end{array}\right]\;$$

Of particular note here are the homogeneous coordinate transformation matrix ${}^{0}A_{1}^{\prime }$ of the coordinate system ${\left(xyz\right)}_{1}$ relative to ${\left(xyz\right)}_{0}$ and ${}^{5}A_{6}^{\prime }$ of the coordinate system ${\left(xyz\right)}_{5}$ relative to ${\left(xyz\right)}_{6}$, which contain 4-DOF variables to transfer the boundary of Double Porro prisms. These are expressed as follows:(19)$${}^{1}A_{0}^{\prime }=T_{2}{}^{1}A_{0}\;$$
(20)$${}^{6}A_{5}^{\prime }=T_{2}{}^{6}A_{5}\;$$

The incident point ${}^{1}P_{1}$ and the refracted ray ${}^{1}\ell_{1}$ on the first virtual boundary, which are relative to coordinate ${\left(xyz\right)}_{1}$, could be calculated by using Equation (4) to Equations (6), (7) and (19). The same method mentioned in the last section is used sequentially to obtain the incident point and incident unit direct vector on each optical boundary. Finally, the spot coordinates on PSD 1 and PSD 2 could be obtained as simultaneous higher order equations in four variables as follows:(21)$$P_{PSD1}=\left[\begin{array}{cccc} X_{1}\left(\theta_{x}{}^{\prime },\theta_{y}{}^{\prime },\delta_{x}{}^{\prime },\delta_{y}{}^{\prime }\right) & Y_{1}\left(\theta_{x}{}^{\prime },\theta_{y}{}^{\prime },\delta_{x}{}^{\prime },\delta_{y}{}^{\prime }\right) & 0 & 1 \end{array}\right]\;$$
(22)$$P_{PSD2}=\left[\begin{array}{cccc} X_{2}\left(\theta_{x}{}^{\prime },\theta_{y}{}^{\prime },\delta_{x}{}^{\prime },\delta_{y}{}^{\prime }\right) & Y_{2}\left(\theta_{x}{}^{\prime },\theta_{y}{}^{\prime },\delta_{x}{}^{\prime },\delta_{y}{}^{\prime }\right) & 0 & 1 \end{array}\right]\;$$

The relationship between PSD 1, PSD 2 and 4-DOF motion of Double Porro prisms is built as Equations (21) and (22), where, $X_{1}$, $X_{2}$, $Y_{1}$ and $Y_{2}$ are the coordinate values of *X* and *Y* direction on PSD 1 and PSD 2, respectively. Consequently, the four laser drift errors could be obtained after solving Equations (21) and (22):(23)$$\theta_{x}{}^{\prime }=f_{1}\left(X_{1},X_{2},Y_{1},Y_{2}\right)\;$$
(24)$$\theta_{y}{}^{\prime }=f_{2}\left(X_{1},X_{2},Y_{1},Y_{2}\right)\;$$
(25)$$\delta_{x}{}^{\prime }=f_{3}\left(X_{1},X_{2},Y_{1},Y_{2}\right)\;$$
(26)$$\delta_{y}{}^{\prime }=f_{4}\left(X_{1},X_{2},Y_{1},Y_{2}\right)\;$$

## 4. Experimental Setup and Results

The prototype of the proposed FSM compensation system is set on the optical bench as shown in Figure 10. As can be seen, this experiment involves the use of a He-Ne laser (EL01A, 632 nm, 10 mW, LASOS, Jena, Germany), a 2-axis linear translation stage, a 2-axis rotation stage, a “Stewart Platform” type hexapod (HXP50-MECA, Newport, Irvine, CA, USA), a double Porro prims composed of two hollow-roof prism mirrors, a beam splitter and two dual-axis lateral PSDs (SPOTANA-9S-USB-L, DUMA OPTRONICS, Nesher, Israel). A He-Ne laser is used as the light source, which is firstly set on the 2-axis linear translation stage. 

Following this, the laser source incidents to a mirror which is set on the 2-axis rotation stage and then reflects to the Double Porro prisms which are set on a 6-axis hexapod. The laser beam is then directed into a beam splitter (BS). Finally, the laser beam is split into PSD 1 and PSD 2 by the BS. The 2-axis linear translation stage and 2-axis rotation stage are applied to produce 4-DOF laser errors, namely $\theta_{x}$, $\theta_{y}$, $\delta_{x}$ and $\delta_{y}$, at the same time. The 6-axis Hexapod is a parallel kinematic motion device that provides 6-DOF motion, which is applied to produce 4-DOF motion ${\theta^{\prime }}_{x}$, ${\theta^{\prime }}_{y}$, ${\delta^{\prime }}_{x}$ and ${\delta^{\prime }}_{y}$ to compensate for the laser errors. The progress of the experiment is as follows. In the beginning, 4 DOF laser errors, $\theta_{x}$, $\theta_{y}$, $\delta_{x}$ and $\delta_{y}$, are produced by the linear translation stage separately, following which PSD 1 and PSD 2 detect the signals of the laser spots. Moreover, a commercial software, MATLAB, is used to solve the laser errors and the compensation values through the reading of two PSDs. Lastly, the Double Porro prisms are driven to compensate for the 4 DOF laser errors. 

Figure 11a,b show the experimental results for the variation of positions of light spots on the PSDs with and without the compensation, when the laser error is $\delta_{x}$ = 50 and $\delta_{x}$ = −50 μm, respectively. As shown in Figure 11a, the readings of PSD1 and PSD are −50 and 50 μm when $\delta_{x}$ = 50 μm. After compensating for the laser error, the readings of PSD 1 and PSD 2 are reduced to approximately 0 μm, which means that the laser error of $\delta_{x}$ = 50 μm is compensated for by the proposed FSM compensation system. The same is seen in Figure 11b, where the result shows that the readings of two PSDs are reduced to around 0 μm, which means that the laser error of $\delta_{x}$= −50 μm is compensated for by the proposed FSM compensation system. Figure 12a,b show the experimental results for the variation of positions of light spots on the PSDs with and without the compensation, when the laser error is $\delta_{y}$ = 50 and $\delta_{y}$ = −50 μm, respectively. As shown, both the laser errors of $\delta_{y}$ = 50 and $\delta_{y}$ = −50 μm are compensated for by the proposed FSM compensation system. Figure 13a,b show the experimental results for the variation of positions of light spots on the PSDs with and without the compensation, when the laser error is $\theta_{x}$ = 0.1 and $\theta_{x}$ = −0.1°, respectively. As shown, both the laser errors of $\theta_{x}$ = 0.1 and $\theta_{x}$ = −0.1 degrees are compensated for by the proposed FSM compensation system. Figure 14a,b show the experimental results for the variation of positions of light spots on the PSDs with and without the compensation, when the laser error is $\theta_{y}$ = 0.1 and $\theta_{y}$ = −0.1°, respectively. As shown, both the laser errors of $\theta_{y}$ = 0.1 and $\theta_{y}$ = −0.1° are compensated for by the proposed FSM compensation system. Figure 15 shows the experimental results for the variation of positions of light spots on the PSDs with and without the compensation when the laser error has 4 DOF errors of $\theta_{x}$ = −0.1°, $\theta_{y}$ = −0.1°, $\delta_{x}$ = −50 μm and $\delta_{y}$ = −50 μm. As shown, 4 DOF laser errors are compensated for by the proposed FSM compensation system. It demonstrates the feasibility of the proposed FSM compensation system.

## 5. Error Analysis

The mathematical modelling of the proposed FSM compensation system is established by using a skew-ray tracing method. However, the real optics system is not exactly the same as the mathematical model due to installation errors of each optical element. The installation errors affect the reading of the PSD and reduce the optical accuracy directly. Therefore, it is essential to analyse the influence of each installation error for the optical elements. 

For the purpose of installation error analysis, each optical element is provided with 6 DOF installation errors respectively in order to observe the influence of installation error on the FSM compensation system; moreover, the coordinate system is shown in Figure 9. Figure 16, Figure 17, Figure 18 and Figure 19 show the installation error analysis of double Porro prisms, BS, PSD 1 and PSD 2 when laser errors are $\delta_{x}$ = 1 mm, $\delta_{y}$ = 1 mm, $\theta_{x}$ = −1° and $\theta_{y}$ = −1°. 

The translational installation errors (${}^{\prime }\delta_{x}$, ${}^{\prime }\delta_{y}$ and ${}^{\prime }\delta_{z}$) and the rotational installation errors (${}^{\prime }\theta_{x}$
${}^{\prime }\theta_{y}$ and ${}^{\prime }\theta_{z}$) are defined as −0.5 mm to −0.5 mm and −0.5° to −0.5°. As shown in Figure 16, Figure 17, Figure 18 and Figure 19, the installation errors of ${}^{\prime }\delta_{z}$, ${}^{\prime }\theta_{x}$, ${}^{\prime }\theta_{y}$ and ${}^{\prime }\theta_{z}$ of each optical element affect the accuracy of the laser error measurement, but ${}^{\prime }\delta_{x}$ and ${}^{\prime }\delta_{y}$ do not. Furthermore, the result of installation error analysis shows that the installation error ${}^{\prime }\theta_{z}$ of PSD 2 is critical in the FSM compensation system due to the fact that it has the longest optical length. 

## 6. Conclusions

This study has presented a new FSM compensation system with Double Porro prisms to compensate for the 4 DOF laser errors of the laser source, which is characterised by shorter optical path length, fewer elements and easier set up at different locations. The performance of the proposed FSM compensation system has been evaluated using a laboratory-built prototype. The experiment results show that the proposed FSM compensation system can eliminate 97% of the laser errors. This implies that the proposed FSM compensation system has the ability to measure and compensate for laser errors.

## 7. Patents

The authors have not published any patent from the work reported in this manuscript.

## Figures and Tables

**Figure 1 sensors-18-04046-f001:**
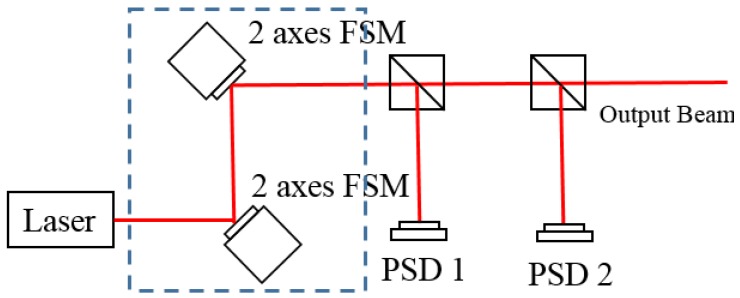
Optical path in conventional FSM compensation system.

**Figure 2 sensors-18-04046-f002:**
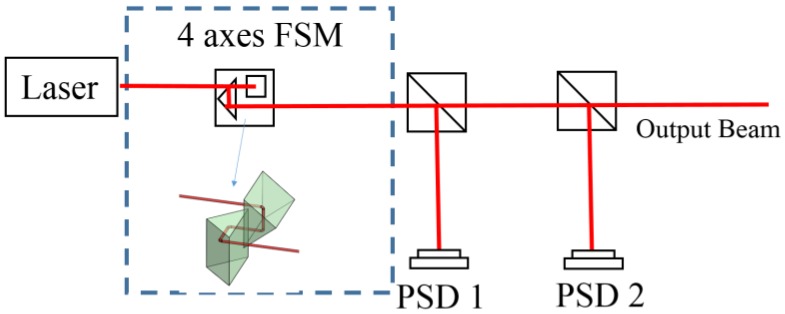
Optical path in proposed FSM compensation system.

**Figure 3 sensors-18-04046-f003:**
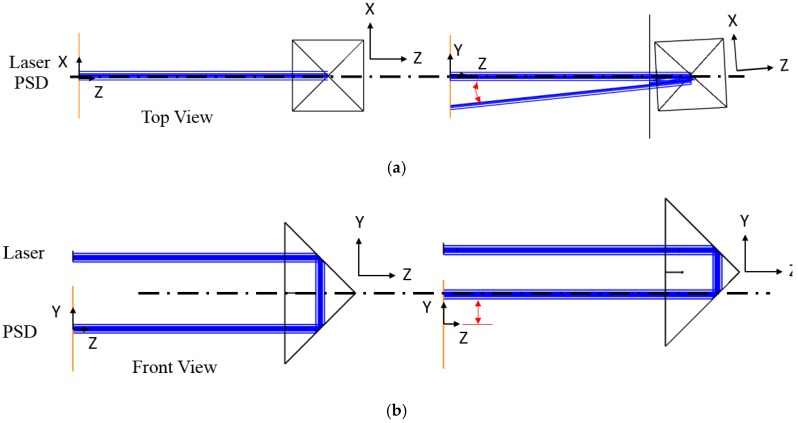
The schematic diagram of (**a**) the angular beam steering and (**b**) the translation beam steering of 90° prisms laser steering.

**Figure 4 sensors-18-04046-f004:**
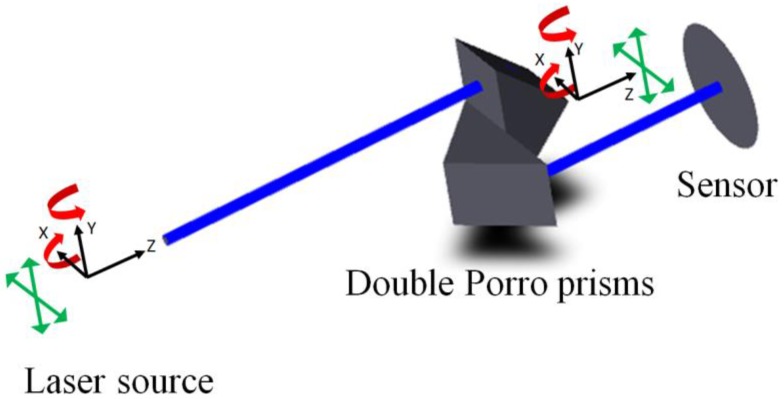
Compensation of four DOF laser errors in proposed FSM compensation system.

**Figure 5 sensors-18-04046-f005:**
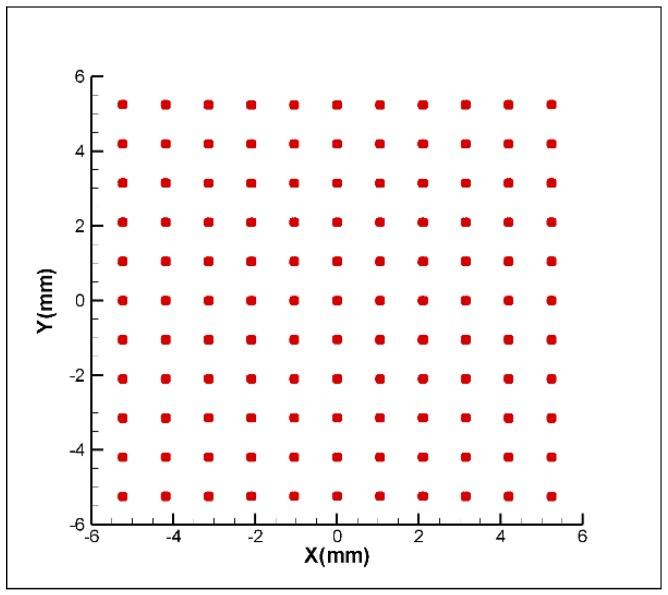
The distributions of laser points without compensation.

**Figure 6 sensors-18-04046-f006:**
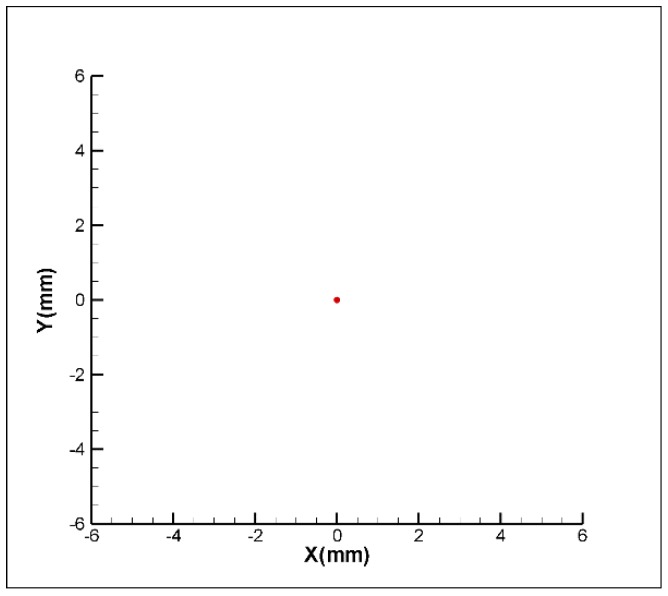
The distributions of laser points with compensation.

**Figure 7 sensors-18-04046-f007:**
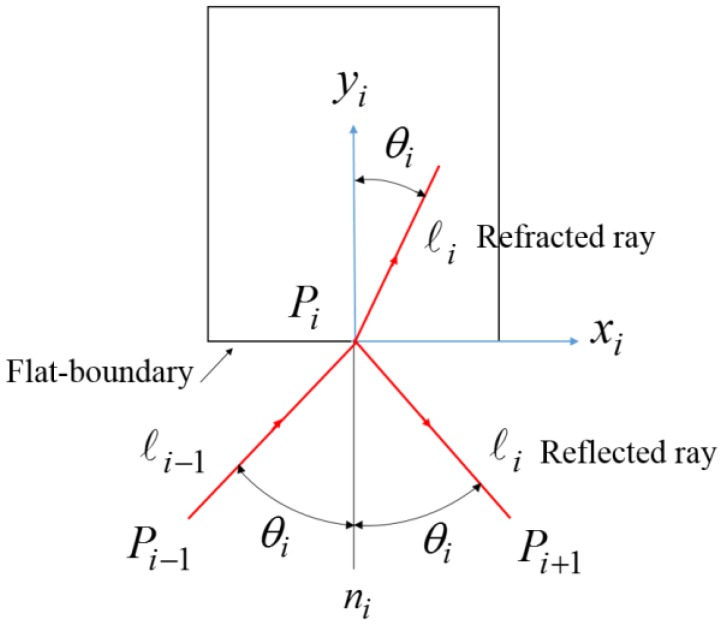
The schematic of refraction and reflection ray.

**Figure 8 sensors-18-04046-f008:**
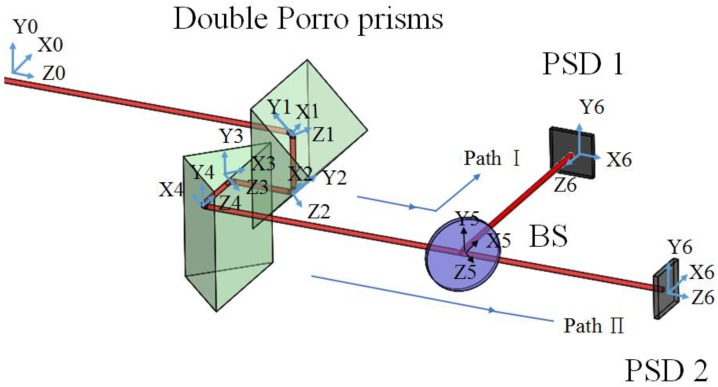
Spatial geometric configuration for Laser drift errors measurement.

**Figure 9 sensors-18-04046-f009:**
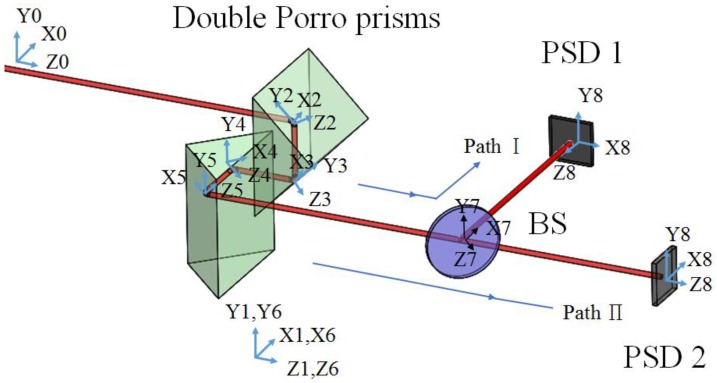
Spatial geometric configuration for Laser drift errors compensation.

**Figure 10 sensors-18-04046-f010:**
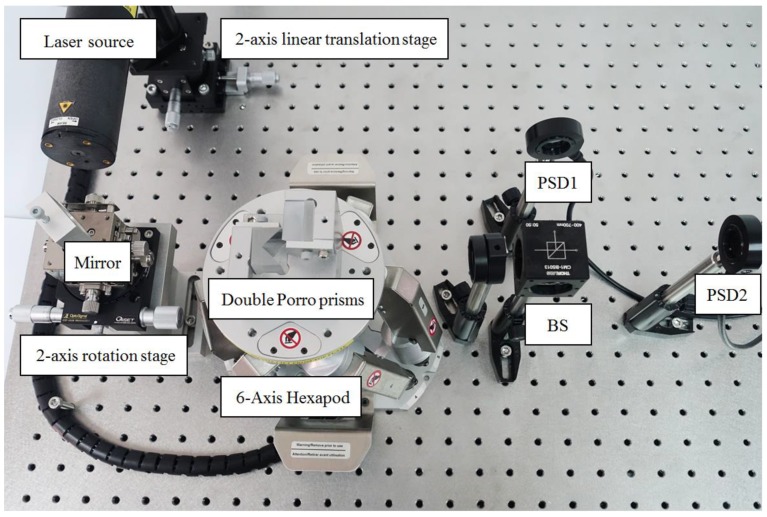
Photograph of experimental setup.

**Figure 11 sensors-18-04046-f011:**
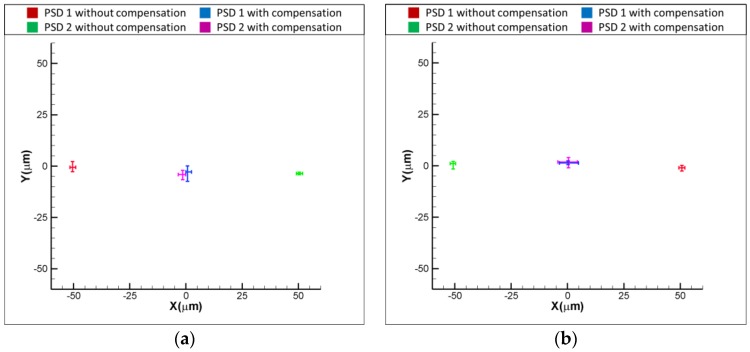
Experimental results for variation of positions of light spots on PSDs with and without compensation (**a**) $\delta_{x}$ = 50 and (**b**) $\delta_{x}$ = −50 μm, respectively.

**Figure 12 sensors-18-04046-f012:**
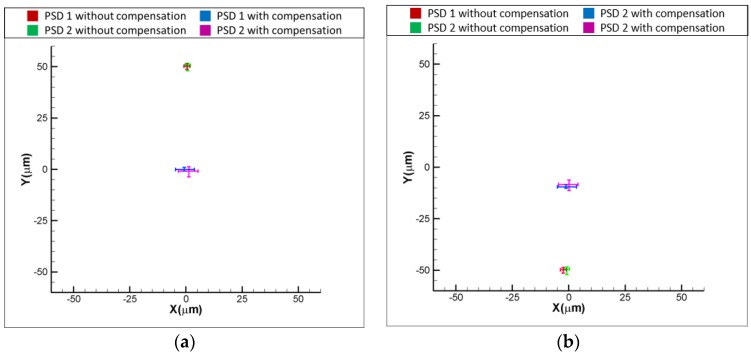
Experimental results for variation of positions of light spots on PSDs with and without compensation (**a**) $\delta_{y}$ = 50 and (**b**) $\delta_{y}$ = −50 μm, respectively.

**Figure 13 sensors-18-04046-f013:**
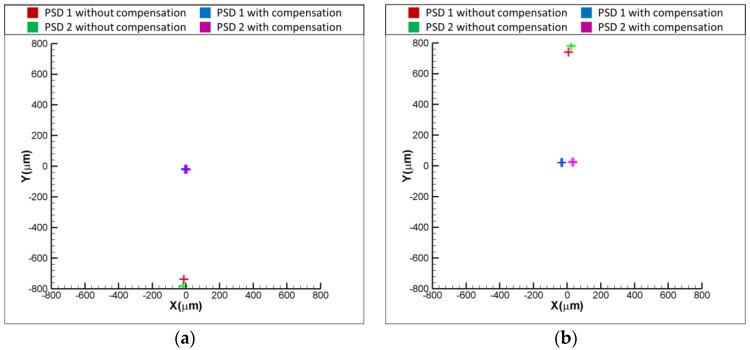
Experimental results for variation of positions of light spots on PSDs with and without compensation (**a**) $\theta_{x}$ = 0.1 and (**b**) $\theta_{x}$ = −0.1°, respectively.

**Figure 14 sensors-18-04046-f014:**
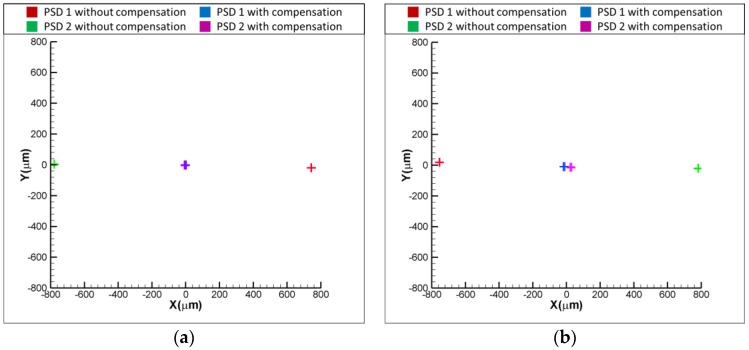
Experimental results for variation of positions of light spots on PSDs with and without compensation (**a**) $\theta_{y}$ = 0.1 and (**b**) $\theta_{y}$ = −0.1°, respectively.

**Figure 15 sensors-18-04046-f015:**
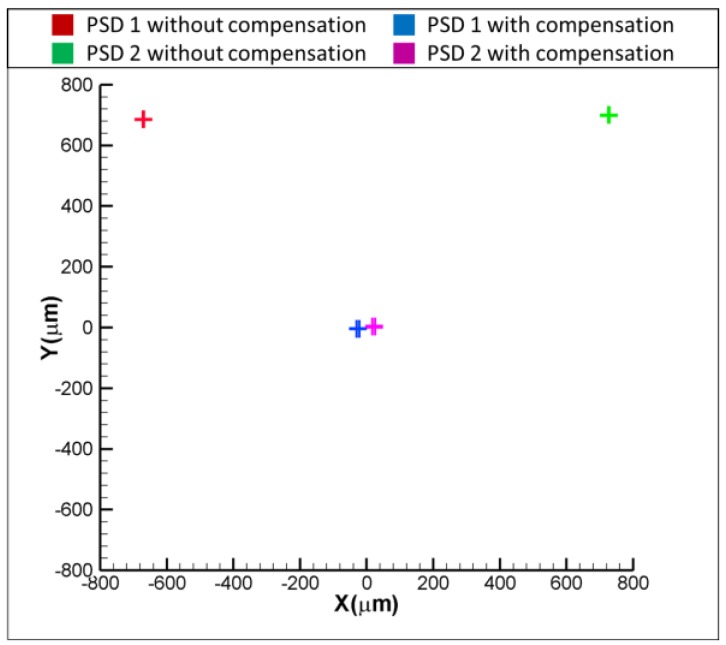
Experimental results for variation of positions of light spots on PSDs with and without compensation when $\theta_{x}$ = −0.1° , $\theta_{y}$ = −0.1°, $\delta_{x}$ = −50 μm and $\delta_{y}$ = −50 μm.

**Figure 16 sensors-18-04046-f016:**
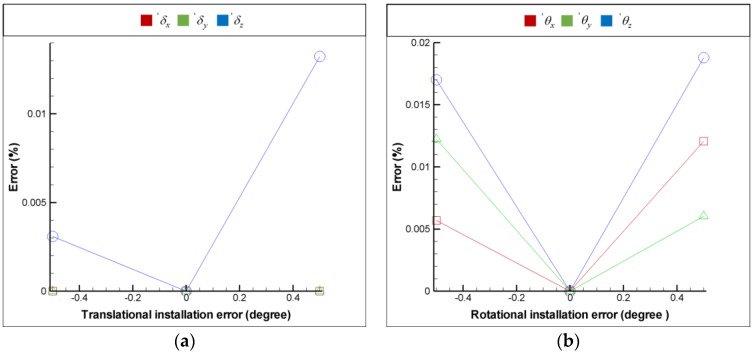
Installation error analysis of double Porro prisms for (**a**) translational installation errors (${}^{\prime }\delta_{x}$, ${}^{\prime }\delta_{y}$ and ${}^{\prime }\delta_{z}$) and (**b**) translational installation errors (${}^{\prime }\theta_{x}$, ${}^{\prime }\theta_{y}$ and ${}^{\prime }\theta_{z}$).

**Figure 17 sensors-18-04046-f017:**
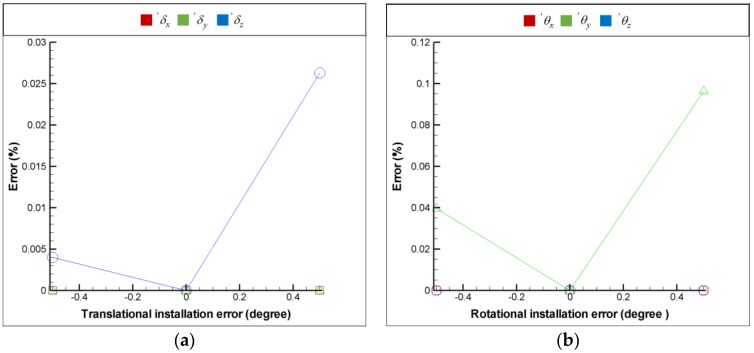
Installation error analysis of double BS for (**a**) translational installation errors (${}^{\prime }\delta_{x}$, ${}^{\prime }\delta_{y}$ and ${}^{\prime }\delta_{z}$) and (**b**) translational installation errors (${}^{\prime }\theta_{x}$, ${}^{\prime }\theta_{y}$ and ${}^{\prime }\theta_{z}$).

**Figure 18 sensors-18-04046-f018:**
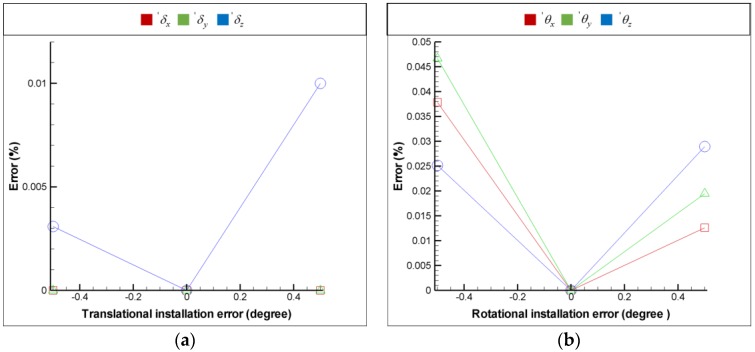
Installation error analysis of double PSD 1 for (**a**) translational installation errors (${}^{\prime }\delta_{x}$, ${}^{\prime }\delta_{y}$ and ${}^{\prime }\delta_{z}$) and (**b**) translational installation errors (${}^{\prime }\theta_{x}$, ${}^{\prime }\theta_{y}$ and ${}^{\prime }\theta_{z}$).

**Figure 19 sensors-18-04046-f019:**
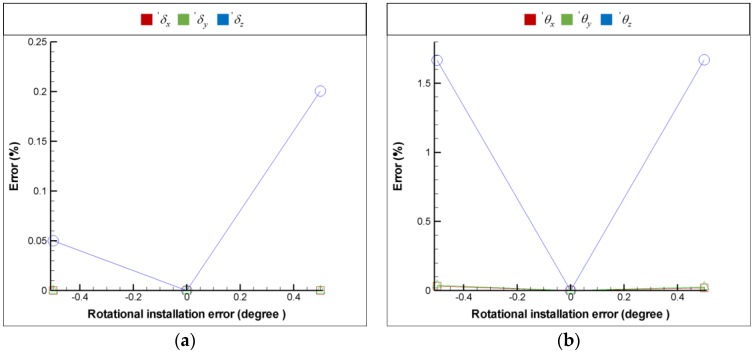
Installation error analysis of double PSD 2 for (**a**) translational installation errors (${}^{\prime }\delta_{x}$, ${}^{\prime }\delta_{y}$ and ${}^{\prime }\delta_{z}$) and (**b**) translational installation errors (${}^{\prime }\theta_{x}$, ${}^{\prime }\theta_{y}$ and ${}^{\prime }\theta_{z}$).

**Table 1 sensors-18-04046-t001:** Coordinate transformation matrix parameters of laser drift errors measurement.

*i*	*i* = 1	*i* = 2	*i* = 3	*i* = 4	*i* = 5	*i* = 6	*i* = 6
Optical Element	Double Porro Prisms	Double Porro Prisms	Double Porro Prisms	Double Porro Prisms	BS	PSD 1	PSD 2
$I_{ix}$	1	1	$1/\sqrt{2}$	0	$1/\sqrt{2}$	$1/\sqrt{2}$	$1/\sqrt{2}$
$I_{iy}$	0	0	0	0	0	0	0
$I_{iz}$	0	0	$-1/\sqrt{2}$	1	$-1/\sqrt{2}$	$-1/\sqrt{2}$	$1/\sqrt{2}$
$J_{ix}$	0	0	1/2	0	0	0	0
$J_{iy}$	$1/\sqrt{2}$	0	$1/\sqrt{2}$	1	1	1	1
$J_{iz}$	$1/\sqrt{2}$	−1	1/2	0	0	0	0
$K_{ix}$	0	0	1/2	−1	$1/\sqrt{2}$	$1/\sqrt{2}$	$1/\sqrt{2}$
$K_{iy}$	$-1/\sqrt{2}$	1	$-1/\sqrt{2}$	0	0	0	0
$K_{iz}$	$1/\sqrt{2}$	0	1/2	0	$1/\sqrt{2}$	$1/\sqrt{2}$	$1/\sqrt{2}$
$t_{ix}$	$L2_{x}$	$L3_{x}$	$L4_{x}$	$L5_{x}$	$L7_{x}$	$L8_{x}$	$L8_{x}$
$t_{iy}$	$L2_{y}$	$L3_{y}$	$L4_{y}$	$L5_{y}$	$L7_{y}$	$L8_{y}$	$L8_{y}$
$t_{iz}$	$L2_{z}$	$L3_{z}$	$L4_{z}$	$L5_{z}$	$L7_{z}$	$L8_{z}$	$L8_{z}$

**Table 2 sensors-18-04046-t002:** Coordinate transformation matrix parameters of laser drift errors compensation.

*i*	*i* = 1	*i* = 2	*i* = 3	*i* = 4	*i* = 5	*i* = 6	*i* = 7	*i* = 8	*i* = 8
Optical Element	4-Axes Actuator	Double Porro Prisms	Double Porro Prisms	Double Porro Prisms	Double Porro Prisms	4-Axes Actuator	BS	PSD 1	PSD 2
Ni	1	reflected	reflected	reflected	reflected	1	reflected		
$I_{ix}$	1	1	1	$1/\sqrt{2}$	0	$1/\sqrt{2}$	$1/\sqrt{2}$	$1/\sqrt{2}$	$1/\sqrt{2}$
$I_{iy}$	0	0	0	0	0	0	0	0	0
$I_{iz}$	0	0	0	$-1/\sqrt{2}$	1	$-1/\sqrt{2}$	$-1/\sqrt{2}$	$-1/\sqrt{2}$	$1/\sqrt{2}$
$J_{ix}$	0	0	0	1/2	0	0	0	0	0
$J_{iy}$	1	$1/\sqrt{2}$	0	$1/\sqrt{2}$	1	1	1	1	1
$J_{iz}$	0	$1/\sqrt{2}$	−1	1/2	0	0	0	0	0
$K_{ix}$	0	0	0	1/2	−1	$1/\sqrt{2}$	$1/\sqrt{2}$	$1/\sqrt{2}$	$1/\sqrt{2}$
$K_{iy}$	0	$-1/\sqrt{2}$	1	$-1/\sqrt{2}$	0	0	0	0	0
$K_{iz}$	1	$1/\sqrt{2}$	0	1/2	0	$1/\sqrt{2}$	$1/\sqrt{2}$	$1/\sqrt{2}$	$1/\sqrt{2}$
$t_{ix}$	$L1_{x}$	$L2_{x}$	$L3_{x}$	$L4_{x}$	$L5_{x}$	$L6_{x}$	$L7_{x}$	$L8_{x}$	$L8_{x}$
$t_{iy}$	$L1_{y}$	$L2_{y}$	$L3_{y}$	$L4_{y}$	$L5_{y}$	$L6_{y}$	$L7_{y}$	$L8_{y}$	$L8_{y}$
$t_{iz}$	$L1_{z}$	$L2_{z}$	$L3_{z}$	$L4_{z}$	$L5_{z}$	$L6_{z}$	$L7_{z}$	$L8_{z}$	$L8_{z}$

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
