# Peer review of "Design and Characterisation of a Fast Steering Mirror Compensation System Based on Double Porro Prisms by a Screw-Ray Tracing Method"

_sensors, 2018, doi:10.3390/s18114046_

Reviewer 1 Report

The use of Porro prism for this purpose is essentially a good idea. The further treatment is rather thorough but I wonder if that is necessary; at least it works as expected.

Some detailed comments:

Title: Design and characterization of a Fast Steering Mirror compensation system based on Porro prisms

4-degree: 4 degrees of freedom (DOF)

equation (1): explain I, J, K, t

equation (6) and (7); maybe there are some "T" superscripts too many or too few, please check

Main problem is the use of the English language: there are many crippled sentences that should be rectified by a thorough editing by an author with an extensive experience in publishing in English.

Author Response

Please see the report. Thank you.

Reviewer 2 Report

This paper presents the Design and Characterization of Novel FSM Compensation System with Porro prisms.
It is an interesting paper; however, I have some comments written below.

In the title avoid the use of an acronym FSM, and I suggest to include screw-ray optics, as this theory discussion takes 6 pages of the paper.

The introduction is OK but the English should be improved.

Design of Proposed FSM compensation system and Simulations
This section needs improvements:
Link xyz in fig. 3 with xyz on fig 4
Porro prisms should properly be called double Porro prisms.
The Zemax simulation is COMPLETELY undocumented – please provide information of the simulation details and the results obtained including errors.

Proposed FSM compensation method
This is a very theoretically challenging section if you are not an expert in screw-ray tracing including homogeneous coordinates and transformation.
Why not start referring the book by Psang Dain Lin, Advanced Geometrical Optics, 2017.
This book introduces the used nomenclature and the underlying theory (Author is also from the same university/Dept.!). The book also includes a chapter about double Porro prisms (ch. 9.7.1) with a lot of the same math described as in this paper!
I also miss a symbol list.
In both table 1 and 2 the number 0.5 is used instead of ½, why?
In general, I find most of the theory presented in this section relevant for an appendix.

Experimental setup and results:
Provide more details of the used components shown on fig. 8, hexapod type, PSD type, Porro prisms type etc..
Figures 9-13 nicely shows the performance of the tested system, however the author’s needs to provide details of the measurement errors including a discussion of which components are most critical.

Conclusion
In the conclusion discuss the performance of the tested system including error analysis instead of just writing “can compensate them well…”

Author Response

Please see the report. Thank you.

Round  2

Reviewer 2 Report

This revised version has improved a lot, and I now find the paper ready for publication.